# AI-Augmented Quantitative MRI Predicts Spontaneous Intracranial Hypotension

**DOI:** 10.3390/diagnostics15182339

**Published:** 2025-09-15

**Authors:** Yi-Jhe Huang, Jyh-Wen Chai, Wen-Hsien Chen, Hung-Chieh Chen, Da-Chuan Cheng

**Affiliations:** 1Graduate Institute of Biomedical Sciences, China Medical University, Taichung 404328, Taiwan; yijhe.huang@msa.hinet.net; 2Department of Radiology, Taichung Veterans General Hospital, Taichung 407219, Taiwan; hubt@gmail.com (J.-W.C.); chenws.tw@gmail.com (W.-H.C.); hungchiehchen@gmail.com (H.-C.C.); 3Department of Post-Baccalaureate Medicine, College of Medicine, National Chung Hsing University, Taichung 402202, Taiwan; 4College of Medicine, National Yang-Ming Chiao Tung University, Taipei 112304, Taiwan; 5Department of Biomedical Imaging and Radiological Science, China Medical University, Taichung 404328, Taiwan

**Keywords:** spontaneous intracranial hypotension (SIH), epidural blood patch (EBP), artificial intelligence (AI), cerebrospinal fluid (CSF) flow, phase-contrast MRI, quantitative imaging

## Abstract

**Background/Objectives**: Spontaneous intracranial hypotension (SIH), caused by spinal cerebrospinal fluid (CSF) leakage, commonly presents with orthostatic headache and CSF hypovolemia. While CSF dynamics in the cerebral aqueduct are well studied, alterations in spinal CSF flow remain less defined. We aimed to quantitatively assess spinal CSF flow at C2 using phase-contrast (PC) MRI enhanced by artificial intelligence (AI) and to evaluate its utility for diagnosing SIH and predicting responses to epidural blood patch (EBP). **Methods**: We enrolled 31 patients with MRI-confirmed SIH and 26 age- and sex-matched healthy volunteers (HVs). All participants underwent ECG-gated cine PC-MRI at the C2 level and whole-spine MR myelography. AI-based segmentation using YOLOv4 and a pulsatility-based algorithm was used to extract quantitative CSF flow metrics. Between-group comparisons were analyzed using Mann–Whitney U tests, and receiver operating characteristic (ROC) analysis was used to evaluate diagnostic and predictive performance. **Results**: Compared to HVs, SIH patients showed significantly reduced CSF flow parameters across all metrics, including upward/downward mean flow, peak flow, total flow per cycle, and absolute stroke volume (all *p* < 0.001). ROC analysis revealed excellent diagnostic accuracy for multiple parameters, particularly downward peak flow (AUC = 0.844) and summation of peak flow (AUC = 0.841). Importantly, baseline CSF flow metrics significantly distinguished patients who required one versus multiple epidural blood patches (EBPs) (all *p* < 0.001). ROC analysis demonstrated that several parameters achieved near-perfect to perfect accuracy in predicting EBP success, with AUCs up to 1.0 and 100% sensitivity/specificity. **Conclusions**: AI-enhanced PC-MRI enables the robust, quantitative evaluation of spinal CSF dynamics in SIH. These flow metrics not only differentiate SIH patients from healthy individuals but also predict response to EBP treatment with high accuracy. Quantitative CSF flow analysis may support both diagnosis and personalized treatment planning in SIH.

## 1. Introduction

Spontaneous intracranial hypotension (SIH) is a debilitating neurological disorder caused by cerebrospinal fluid (CSF) leakage, often presenting with orthostatic headache and impairing daily function. Epidural blood patching (EBP) is the most commonly employed treatment, but its therapeutic response varies significantly. While EBP is the mainstay, predicting its success remains challenging, and conventional imaging findings for SIH diagnosis can be subtle or inconsistent [1,2].

Therefore, there is a pressing need for more objective and precise diagnostic tools for SIH, especially given that conventional imaging can be inconclusive. Recent advances in imaging techniques, such as the development of probabilistic scoring systems leveraging intracranial MRI findings [3], aim to improve the early diagnosis of SIH and the detection of CSF leaks, thereby facilitating timely intervention with therapies like epidural blood patch (EBP).

Given that CSF leakage is the core pathophysiology of SIH, changes in CSF flow dynamics are hypothesized to be early and sensitive indicators, potentially enabling earlier disease diagnosis. Advances in neuroimaging have facilitated the assessment of CSF dynamics, particularly through phase-contrast magnetic resonance imaging (PC-MRI), which allows for non-invasive, quantitative measurement of CSF flow. Notably, most existing PC-MRI studies have focused on the cerebral aqueduct, where SIH patients typically demonstrate reduced CSF flow during the disease state [4] and progressive recovery following effective treatments such as EBP [5]. These results suggest that aqueduct-based CSF flow could serve as a potential diagnostic and treatment-monitoring biomarker.

Intracranial aqueduct cine PC-MRI has demonstrated value in distinguishing SIH from healthy states and tracking treatment response [4,5]. As most CSF leaks in SIH arise from the spine, we asked whether site-proximal cervical (C2-3) flow metrics could also reflect disease status and severity. Evidence on cervical CSF flow, however, remains limited and inconsistent across studies [6,7,8,9] (Appendix A). Wolf et al., (2024) reported increased CSF velocity and displacement at C2/C3 in SIH patients, suggesting possible compensatory spinal dynamics [7,8]. In contrast, Tsai et al., (2018), using an MR-based intracranial pressure estimation method (MR-ICP), demonstrated significantly reduced cervical CSF flow in SIH patients [6]. These conflicting results underscore the need for more objective and robust methods for region-specific analysis of CSF dynamics at the spinal level, particularly for diagnostic purposes. Manual measurement of CSF flow using PC-MRI is prone to inter-observer variability and measurement errors, especially in the cervical spine where CSF flow can be non-uniform and complicated by the presence of enlarged epidural venous plexuses. This inherent subjectivity and potential for small errors in manual segmentation highlight the need for automated approaches. We therefore developed an automated, region-specific pipeline to quantify spinal CSF dynamics.

In recent years, artificial intelligence (AI) has shown significant promise in automating medical image analysis, or segmentation to enhance accuracy, reproducibility, and efficiency. With the widespread adoption of deep learning, cerebrospinal fluid (CSF) segmentation has rapidly transitioned from traditional thresholding, clustering, and handcrafted feature-based workflows toward end-to-end semantic segmentation centered on convolutional neural networks (CNNs). In brain imaging with MRI and CT, encoder–decoder architectures such as U-Net variants and transfer learning from pre-trained models (e.g., VGG16) are most commonly employed. Reported Dice coefficients often exceed 0.8 in clinical datasets, with applicability across both pediatric and adult populations [10,11,12]. Beyond CNNs, multiple classifier systems—typically ensembles of multilayer perceptrons (MLPs) fused via majority voting—can enhance robustness in cases with noise or intensity inhomogeneity [13]. Classical unsupervised clustering approaches (e.g., k-means, expectation–maximization) and adaptive thresholding methods—often combined with anatomical priors and connectivity analysis—are computational efficiency; however, their accuracy in complex pathological scenarios generally lags behind deep learning–based techniques [14,15]. More recently, studies using automated CSF and intracranial volumetry on emergent head CT have demonstrated the feasibility of real-time clinical applications for assessing cerebral edema and mass effect [12]. Spinal imaging presents additional challenges due to smaller anatomical scales, lower contrast, and frequent motion or susceptibility artifacts, necessitating more specialized network designs and structural constraints. CNN-based specialized architectures, such as multi-path dense networks and U-Net variants, have demonstrated strong performance in multi-center, multi-parametric MRI, enabling simultaneous segmentation of CSF, spinal cord, and surrounding structures while preserving boundary details [16,17]. Atlas-based methods with topology constraints, including deformable atlases combined with topology-preserving mechanisms, improve anatomical plausibility across varying fields of view and imaging modalities [18]. In contrast, classical clustering and fuzzy methods are generally less effective than in brain imaging and often require anatomical priors or post-processing steps to ensure structural continuity [17,18,19]. Deep learning models such as YOLO (You Only Look Once) and U-Net++ can detect and segment CSF-containing regions, enabling objective quantification. Fu et al. (2022) further demonstrated that a cascade AI model integrating YOLOv3 and U-Net++ can accurately quantify spinal CSF volume in SIH patients, and that CSF volume measurements correlated with clinical recovery, reinforcing the potential of AI-assisted analysis for diagnostic [20].

For CSF flow metrics evaluation, automated AI-based segmentation demonstrates clear advantages as compared with traditional ROI placement methods in PC-MRI of the cerebral aqueduct. Deep learning–based CNN models (e.g., U-Net, MultiResUNet) achieve high concordance with expert neuroradiologist manual delineations for aqueduct segmentation and flow quantification, with Dice coefficients reaching ≥0.93, lower segmentation failure rates, and superior consistency of flow parameters, thereby markedly reducing inter- and intra-operator variability while improving efficiency and reproducibility [21]. Second, semi-automated approaches (e.g., seed-based or semi-automatic contouring methods) help mitigate bias introduced by subjective adjustments, and their flow metrics typically yield significantly higher intraclass correlation coefficients (ICC) compared with the conventional fixed-radius approach [22]. Furthermore, several studies have highlighted that ROI size and placement strongly affect quantitative outcomes such as stroke volume; automated methods adaptively update aqueduct boundaries in accordance with morphological changes across the cardiac cycle, thereby maintaining more robust and accurate measurements under varying cardiac phases [9,22,23,24]. Taken together, AI-driven automated and semi-automated segmentation has emerged as the predominant methodological pathway for PC-MRI CSF flow analysis—particularly in the cerebral aqueduct—offering higher consistency, accuracy, and efficiency than manual ROI placement. (see Appendix A for a side-by-side comparison) While AI has increasingly been applied in diagnostic radiology, its use in objective, quantitative assessment of dynamic CSF flow metrics at the spinal level for SIH diagnosis remains underexplored.

This study aims to bridge that gap by employing AI-augmented quantitative PC-MRI to measure CSF flow dynamics at the C2 spinal level in SIH patients. By comparing these parameters with healthy controls, our primary objective is to investigate the utility of cervical CSF flow metrics as novel, objective imaging biomarkers for the diagnosis of SIH, aiming to reduce dependence on manual, subjective interpretations.

## 2. Materials and Methods

### 2.1. Study Design and Participants

This study used a retrospective case–control design for patients with SIH and prospective recruitment for healthy volunteers (HVs). Consecutive SIH cases were retrospectively identified from clinical and imaging archives between January 2010 and December 2021 under institutional review board (IRB) approvals CE22242B (January 2010 to December 2021) and SE17334A (December 2017 to December 2020). HVs were prospectively consented between July 2016 and October 2017 under CE16103A, and were matched by age and sex to the SIH cohort. All HVs underwent whole-spine MR myelography (MRM) to exclude asymptomatic CSF leaks and ECG-gated cine PC-MRI at the C2 level for flow quantification; conventional whole-spine MRI was also acquired in the same session.

For SIH patients, diagnosis was based on clinical symptoms and MRI findings suggestive of spinal CSF leakage. Complete recovery was defined as the absence of clinical symptoms and normalization of radiological findings on follow-up MRI and MR myelography (MRM). Patients with incomplete imaging coverage, uneven signal intensity, or significant artifacts were excluded due to poor image quality. In the present analysis, 31 SIH patients and 26 HVs met all criteria and were included.

The protocol was approved by the institutional review board (SE17334A, CE16103A, CE22242B). Written informed consent was obtained from healthy volunteer participants. For the retrospectively analyzed clinical cohort, the IRB approved a waiver of informed consent in accordance with local regulations.

### 2.2. Imaging Techniques

At baseline, all SIH patients underwent a comprehensive MRI protocol including conventional brain MRI, whole-spine MRI, whole-spine MR myelography (MRM), and ECG-gated cine PC-MRI at the C2 level. HVs underwent a single session consisting of whole-spine MRI, whole-spine MRM to exclude asymptomatic CSF leaks, and ECG-gated cine PC-MRI at the C2 level for flow quantification. For post-treatment assessment, SIH patients who received EBP underwent a follow-up (post-1st-EBP) MRI within 1–14 days to evaluate early response, which includes whole-spine MR, whole spine MR myelography and ECG-gated cine PC-MRI at the C2 level. The timing of a second EBP, when needed, was determined clinically based on symptoms and imaging. Additional follow-up MRI after a second EBP was obtained on a discretionary basis. Patients treated with hydration alone typically did not undergo repeat inpatient MRI once symptoms resolved.

We used a 1.5 T MRI scanner (MAGNETOM Aera, Siemens Healthcare, Erlangen, Germany) equipped with a 20-channel phased-array head and spine coil to perform all images.

The imaging protocol of PC-MR included sagittal gradient-echo T2*-weighted and 3D axial time-of-flight (TOF) scans for localization, followed by three consecutive ECG-gated PC-MRI sequences with low velocity encoding (venc = 10–15 cm/s) to assess CSF flow. Thirty-two cardiac phases were acquired per sequence with the following parameters: repetition time (TR) = 120 ms, echo time (TE) = 10 ms, flip angle = 10°, slice thickness = 5 mm, field of view = 14–16 cm, matrix = 256 × 256, and two signal averages. Each scan lasted approximately 90–100 s, totaling around 15 min for all three sequences. CSF flow was measured at the mid-C2 vertebral body level, where the subarachnoid space has a consistent cylindrical shape.

In addition, SIH patients underwent conventional brain MRI at baseline, including axial spin-echo T1-weighted images (TR/TE = 500/10), axial fast spin-echo T2-weighted images (TR/TE = 3200/115), and gadolinium-enhanced T1-weighted imaging in axial, sagittal, and coronal planes.

To evaluate CSF leakage sites, whole-spine MRM was performed using a 3D sampling perfection with optimized contrast using different flip-angle evolution (3D-SPACE) sequence for SIH patients at both initial and follow-up stages. The MRM parameters were as follows: TR = 3000 ms, TE = 560 ms, isotropic voxel size = 0.9 mm^3^, matrix = 320 × 320, field of view = 200 mm. Fat suppression and GRAPPA reconstruction (acceleration factor = 2) were applied. Coronal volumetric images were acquired separately from the cervical-to-thoracic and thoracic-to-lumbar spine.

### 2.3. Artificial Intelligence-Based Flow Analysis

For the automated analysis of CSF flow at the C2 level using PC-MRI data, an AI-based algorithms were employed within the Matlab environment (The MathWorks Inc., Natick, MA, USA, (2024). MATLAB version: 24.1.0.2603908 (R2024a)) [25]. Figure 1 shows the schematic workflow of automated CSF analysis, including both SIH patients and HVs. All PC-MRI raw data underwent preliminary processing including image resizing, image format conversion and phase contrast image baseline correction, etc., to ensure consistent input for the following algorithms.

#### CSF Region Detection

In the first step, magnitude images were processed using the YOLOv4 (You Only Look Once version 4) object detection framework [26], which was specifically adapted to identify the CSF spaces within the spinal canal. The objective of this step was to ensure comprehensive coverage of all regions through which CSF flows, while minimizing the inclusion of non-CSF tissues to reduce potential interference in subsequent analyses. Two experienced radiologists manually annotated the ground truth using the MATLAB Image Labeler App. A total of 151 spinal canal MRI images were collected for model training. These images were completely independent from the 31 SIH patients and 26 healthy controls included in the subsequent statistical analyses. The dataset was divided as follows: 60% (90 images) for the training set (used to update YOLO weights), 10% (15 images) for the validation set (to prevent overfitting), and 30% (46 images) for the independent test set (for final performance evaluation).

To improve the generalization of the model, we applied data augmentation during training. Specifically, we implemented random horizontal flipping (with or without), random scaling (90–110%, step 1%), and random rotation (−10° to +10°, step 1 degree) using customized transform functions. These augmentations simulated natural variations en-countered in clinical imaging, such as differences in patient positioning and scan angles, thereby improving the robustness of the model. The pre-trained model employed in this study was Tiny-YOLOv4-COCO, which had been previously trained on the COCO dataset. This enabled the network to acquire generalized low- and mid-level features (such as edges, structures and textures), which efficiently extracted available features even if applied to medical images. Fine-tuning was then performed afterwards using the extra 151 spinal canal MRI images, which were entirely distinct from the 31 SIH patients and 26 healthy controls analyzed in the main study. For the analysis of the 46 images in the independent test set, an average precision (AP) of 0.91 was achieved under the condition that the inter-section over union (IoU) exceeded 0.5.

After detecting the spinal canal region, the analysis proceeded with flow feature extraction by phase images. Independent Component Analysis (ICA) [27] was applied to decompose the mixed flow signals extracted by YOLOv4 within the CSF region for each voxel, under the assumption that the spinal canal region’s signals primarily consist of three components: CSF, blood flow, and static tissue. This process allowed for the identification of independent components representing distinct pulsatile patterns associated with CSF flow. From these components, we filtered out signals with low similarity and removed extraneous noise (as shown in Figure 2a). The preliminary CSF reference velocity waveform was derived by averaging the value of the remaining signals (indicated by the pink asterisk in Figure 2b).

Subsequently, following the experimental approach of Alperin et al. [28] the Pulsatility-Based Segmentation (PUBS) algorithm was used to refine the CSF region segmentation and to characterize its pulsatile flow dynamics. PUBS leverages the inherent pulsatile nature of CSF flow to precisely differentiate it from static tissues and other fluctuating signals, such as those from enlarged epidural venous plexuses. Using the reference curve from ICA result, we analyzed all voxels identified by YOLOv4 within the CSF region. The normalized cross-correlation (NCC) coefficient was employed to assess the similarity of individual voxel velocity waveforms to the reference curve. The NCC values, normalized between −1 and 1, represent the degree of correlation, with values approaching 1 indicating strong positive correlation. By adjusting the NCC threshold, different levels of CSF boundary segmentation were achieved. Based on Alperin’s experimental findings, we determined that the NCC threshold of 0.5 provided optimal segmentation and was used as the final criterion for defining the CSF region. The output of PUBS is a highly refined CSF mask, providing accurate and reproducible CSF flow measurements. A schematic representation of the process described above is presented in Figure 3.

The refined CSF masks generated by the AI algorithm enabled the automatic extraction of quantitative CSF flow parameters. Before calculating the physiological parameters, background phase correction was conducted on the phase images. This correction is essential, as hardware-related imperfections during the MRI acquisition process can cause the phase of static tissues to deviate from zero. Adjusting the phase of these static tissues enhances the accuracy of quantification in other regions. To further minimize physiological variability, each image set was acquired sequentially three times using identical imaging parameters, and the average of these three acquisitions served as the final dataset for subsequent comparisons.

The flow parameters were automatically derived from the results of AI-based algorithms. Considering the directional characteristics of CSF flow, the mean flow was calculated separately for cranial (upward) and caudal (downward) directions, in addition to the total flow regardless of direction. Peak flow was defined as the maximum flow value in each respective direction. Furthermore, the net output per cardiac cycle and the absolute stroke volume were also computed for comprehensive analysis.

### 2.4. Treatment and Follow-Up

Patients diagnosed with SIH who did not respond to conservative hydration therapy underwent EBP. The EBP was administered one or two vertebral levels below the area exhibiting the most significant abnormal CSF signals in the neural sleeves, as identified on magnetic resonance imaging (MRI). All procedures were conducted by an experienced anesthesiologist using a 20-gauge epidural Tuohy needle via a midline approach, with the patient positioned in lateral recumbency.

The epidural space was identified using the loss-of-resistance technique for lumbar injections and the hanging-drop technique for cervical or thoracic levels. Autologous blood was injected until the patient reported symptoms such as headache, back pain, or discomfort, indicating the endpoint of the procedure. Following the EBP, patients were instructed to remain in the supine position for a minimum of two hours.

The efficacy of the initial EBP was primarily assessed through clinical improvement. Further EBP procedures were conducted if the clinical response was deemed inadequate.

For each patient, the vertebral level of the first EBP was recorded and categorized as thoracic, or lumbar; no cervical EBPs were performed (Appendix A).

### 2.5. Statistical Analysis

Data analysis was performed using SPSS (v. 21; SPSS Inc., Chicago, IL, USA) [29]. The normal distribution of continuous variables was tested by the Kolmogorov–Smirnov method. A nonparametric Mann–Whitney test was used to examine non-normally distributed variables. Fisher’s exact test and χ^2^ test were used to analyze nominal variables. For analyses of EBP effectiveness, predictor variables were derived from the post-1st-EBP PC-MRI acquired within 1–14 days after the first EBP. Patients were categorized as first EBP success or first EBP failure (multiple EBP required). The receiver operating characteristic (ROC) curves, cutoffs, and accuracy metrics were thus referenced to this early post-treatment timepoint to stratify the need for repeated EBP. Statistical significance was set at *p* < 0.05. Post-treatment outcomes were analyzed using comparative statistics to correlate CSF flow dynamics with the number of EBPs needed for clinical improvement.

## 3. Results

### 3.1. Participant Characteristics (Table 1)

A total of 31 patients with SIH comprised the test group, while 26 age- and sex-matched HVs formed the control group. No significant differences were observed between the groups in terms of age (39.6 ± 10.0 years vs. 38.1 ± 6.8 years, *p* = 0.724) or sex distribution (female/male: 22/9 vs. 15/11, *p* = 0.575). The median duration from symptom onset to the first MRI in SIH patients was 8 days (range: 2–60). Among the patients (the test group), 25 received EBP treatment, while 6 recovered with hydration alone. A second or subsequent EBP was required in 12 cases. The median interval from the first MRI (baseline) to the initial EBP was 2 days (range: 1–7), and from the first EBP to the follow-up post-1st-EBP MRI was also 2 days (range: 1–14).

### 3.2. Comparison of CSF Flow Parameters at Baseline MR Between SIH Patients and Healthy Volunteers (Table 2)

Upward mean cerebrospinal fluid (CSF) flow (0.76 ± 0.31 vs. 1.18 ± 0.34 mL/s), downward mean CSF flow (1.01 ± 0.43 vs. 1.60 ± 0.54 mL/s), and the total mean flow (1.77 ± 0.72 vs. 2.78 ± 0.84 mL/s) were all significantly lower in the SIH group (*p* < 0.001). Similarly, both upward peak flow (1.28 ± 0.50 vs. 1.87 ± 0.52 mL/s) and downward peak flow (1.76 ± 0.77 vs. 2.96 ± 0.92 mL/s), as well as their total (3.04 ± 1.26 vs. 4.83 ± 1.39 mL/s), were markedly reduced in SIH patients (all *p* < 0.001). Additionally, the total CSF flow per cardiac cycle in both directions and the absolute stroke volume were significantly lower in SIH patients (absolute stroke volume: 27.54 ± 11.03 vs. 42.98 ± 12.86 mL/cycle, *p* < 0.001).

### 3.3. CSF Flow Comparison After Recovery (Table 3)

Of the 31 SIH patients, six patients improved with hydration alone. Among the 25 patients who received an EBP, 13 recovered after a single EBP and 12 required multiple EBPs. One multi-EBP patient had residual epidural fluid accumulation with persistent chronic headache and was therefore not classified as complete recovery. Consequently, Table 3 summarizes CSF flow parameters of the 24 SIH patients achieving complete recovery during our follow-up. Among these 24 SIH patients who underwent follow-up imaging after complete recovery, most CSF flow parameters showed improvement from their baseline SIH values.

### 3.4. SIH Diagnostic Performance of CSF Flow Metrics at Baseline MR (Table 4)

Receiver operating characteristic (ROC) analysis demonstrated that several CSF flow parameters possessed good diagnostic performance in differentiating SIH patients from HVs. The area under the curve (AUC) values for all selected parameters exceeded 0.79, indicating reliable discriminative ability. Among them, downward peak flow (AUC = 0.844) and summation of peak flow (AUC = 0.841) showed the highest diagnostic accuracy, each achieving 92.3% specificity and over 70% sensitivity at their respective cutoff values (2.0964 and 3.5265 mL/s). Similarly, summation of mean flow (AUC = 0.828, cutoff = 1.9812 mL/s), absolute stroke volume (AUC = 0.829, cutoff = 30.8367 mL/cycle), and upward peak flow (AUC = 0.793, cutoff = 1.4647 mL/s) also demonstrated strong classification performance, with sensitivities ranging from 67.7% to 74.2% and specificities from 84.6% to 92.3%. These findings suggest that multiple CSF flow metrics, especially peak flow and stroke volume parameters, may serve as valuable physiological markers for distinguishing SIH from healthy states.

### 3.5. Association Between Post-1st-EBP PC-MRI CSF Flow Parameters and the Number of EBPs

Among SIH patients who received EBP, those with successful outcomes (*n* = 13) after first EBP exhibited significantly higher post-1st-EBP PC-MRI CSF flow parameters across all metrics compared to patients with first EBP failure (*n* = 12) (Table 5). Specifically, upward mean flow (1.26 ± 0.23 vs. 0.69 ± 0.14 mL/s), downward mean flow (1.81 ± 0.46 vs. 0.84 ± 0.24 mL/s), and the total mean flow (3.07 ± 0.65 vs. 1.53 ± 0.36 mL/s) were notably elevated in the successful group (all *p* < 0.001). Similarly, upward peak flow (2.00 ± 0.34 vs. 1.13 ± 0.24 mL/s), downward peak flow (2.98 ± 0.64 vs. 1.45 ± 0.42 mL/s), and their total (4.98 ± 0.95 vs. 2.58 ± 0.61 mL/s) were significantly greater in patients with EBP success (all *p* < 0.001). Additionally, total CSF volume per cardiac cycle—including both upward (24.26 ± 5.13 vs. 12.47 ± 2.64 mL/cycle) and downward flow (22.92 ± 4.35 vs. 14.58 ± 2.95 mL/cycle)—as well as the absolute stroke volume (47.17 ± 9.16 vs. 24.04 ± 5.45 mL/cycle), were all significantly higher in the successful group. These findings suggest that elevated post-1st-EBP MR CSF flow dynamics may predict a favorable response to EBP treatment in patients with SIH.

Regarding the injection level of the first EBP, no cervical EBPs were performed in either group. In the failure group (*n* = 12), EBPs were thoracic in 10 (83.3%) and lumbar in 2 (16.7%); in the success group (*n* = 13), EBPs were thoracic in 9 (69.2%) and lumbar in 4 (30.8%). The distribution of thoracic versus lumbar levels did not differ between groups (Fisher’s exact *p* = 0.64).

### 3.6. Diagnostic Performance of CSF Flow Metrics at Post-1st-EBP MR for Predicting First EBP Effectiveness

ROC analysis based on post-1st-EBP PC-MRI demonstrated excellent discrimination between patients with first EBP success and patients with first EBP failure groups (Table 6). These included downward mean flow (cutoff: 1.2511 mL/s), downward peak flow (2.1968 mL/s), summation of peak flow (3.4623 mL/s), downward CSF total flow (16.4993 mL/cycle), and absolute stroke volume (33.3892 mL/cycle). Other parameters also demonstrated excellent performance, with upward mean flow (AUC = 0.994, cutoff: 0.902 mL/s), summation of mean flow (AUC = 0.994, 2.1521 mL/s), upward peak flow (AUC = 0.994, 1.4923 mL/s), and upward CSF total flow (AUC = 0.974, 17.1383 mL/cycle) each showing 92.3% sensitivity and 100% specificity. These findings indicate that early post-treatment (post-1st-EBP) CSF flow parameters, particularly those reflecting overall flow magnitude, may serve as highly accurate and noninvasive biomarkers for predicting the success of EBP in patients with spontaneous intracranial hypotension.

## 4. Discussion

Our study has yielded several key insights into the pathophysiology, diagnosis, and management of SIH based on quantitative cerebrospinal fluid (CSF) flow metrics derived from PC-MRI.

First, we observed significantly reduced CSF flow dynamics at the C2 level in SIH patients compared to healthy volunteers. As shown in Table 2, SIH patients exhibited lower values across all measured parameters, including upward and downward mean and peak flow, total flow per cardiac cycle, and absolute stroke volume (all *p* < 0.001). These findings indicate a global impairment of spinal CSF dynamics associated with CSF hypovolemia. Importantly, follow-up imaging at complete recovery showed a trend toward normalization of CSF flow metrics in most patients, and the differences compared to HVs were not statistically significant (Table 3). This suggests that CSF flow impairment in SIH may be reversible following successful treatment.

Second, by integrating artificial intelligence (AI)-assisted processing with PC-MRI, we established an efficient and reproducible method for automated CSF flow quantification. ROC curve analysis demonstrated that several flow parameters exhibited strong diagnostic potential for distinguishing SIH patients from HVs (Table 4). Specifically, downward peak flow (AUC = 0.844) and summation of peak flow (AUC = 0.841) displayed high sensitivity and specificity, underscoring their importance as noninvasive physiological biomarkers.

Most importantly, our results provide novel evidence that post-1st-EBP MR flow metrics may serve as critical predictors of treatment response to first EBP. Patients requiring multiple EBPs showed significantly lower post-1st-EBP MR CSF flow across all measured parameters compared to those who recovered after a single EBP (Table 5). Notably, ROC analysis revealed that several parameters—including downward mean flow, peak flow, and stroke volume—achieved perfect diagnostic accuracy (AUC = 1.0) in distinguishing EBP success from failure (Table 6). These findings emphasize the potential utility of quantitative CSF flow assessment in guiding therapeutic decision-making and identifying patients at risk for suboptimal responses, who may benefit from early intervention or escalated treatment strategies.

The findings of CSF flow changes at the upper cervical spinal level (C2/C3) have indeed remained inconsistent across studies in SIH patients, as introduced at the beginning of this manuscript. Previous studies have shown that variations in CSF flow may be influenced by epidural venous plexus dilatation and cervical canal narrowing. For instance, a 2023 Neurology study [7,8] reported increased CSF flow at the cervical spine in some SIH patients, possibly due to venous congestion compressing the subarachnoid space and creating a jet-like effect at narrowed levels.

In contrast, our data predominantly show reduced CSF flow across the cervical spine, supporting a global hypovolemic state in most SIH cases. This finding is largely consistent with earlier research by Tsai et al. (2018) [6], who utilized MRI to assess CSF dynamics and intracranial elastance (IE) in SIH patients. Their study also found a significant decrease in transcranial inflow and outflow for SIH patients overall (*p* < 0.01). Furthermore, and critically, Tsai et al. (2018) [6] differentiated SIH patients based on epidural venous engorgement (EVD) status. They reported that EVD-negative SIH patients exhibited significantly lower IE (0.055 ± 0.012 mmHg/cm/mL) and significantly lower peak-to-peak CSF pressure gradient (PGcsf-pp) (0.024 ± 0.007 mmHg/cm) when compared to both normal volunteers and the EVD-positive SIH group. Conversely, EVD-positive patients demonstrated higher IE and (PGcsf-pp) compared with healthy controls, representing a different direction of change in these parameters. The presence of EVD itself is often considered an important radiological marker and a sign of significant compensatory mechanisms in SIH. EVD-positive patients may thus represent a more severe or chronic presentation, where the expansion of the epidural venous plexus serves as a compensatory mechanism for reduced CSF volume, potentially leading to distinct or more complex clinical and physiological manifestations, including localized alterations in CSF flow as reported by other studies. This nuanced understanding of SIH heterogeneity, particularly concerning EVD status, provides a plausible explanation for why studies like Wolf et al.’s [7,8], focusing on cases with venous congestion, might show increased cervical CSF flow, whereas our results predominantly show reduced cervical CSF flow in most SIH patients. By revisiting this issue raised in the Introduction, our findings thus help reconcile previous inconsistencies and highlight the importance of patient stratification in interpreting CSF flow changes in SIH.

The precise location and strategy of EBP delivery are crucial factors influencing treatment success in spontaneous intracranial hypotension (SIH). Prior studies, such as those by Cheema et al. (2023) [30] and Hazama et al. (2023) [31], have demonstrated the effectiveness of targeted cervical EBP in patients who had previously failed lumbar or blind EBP procedures. In our study, all patients received targeted EBP at the suspected CSF leak site, guided by MRM findings and performed by experienced anesthesiologists using a midline approach.

Intriguingly, our new findings reveal that post-1st-EBP MR CSF flow parameters at the C2 level possess significant predictive power for EBP success. As detailed in Table 5, all measured CSF flow parameters—including upward mean flow, downward mean flow, summation of mean flow, upward peak flow, downward peak flow, summation of peak flow, upward CSF total flow, downward CSF total flow, and absolute stroke volume—were significantly higher in patients who achieved EBP success compared to those who experienced EBP failure (all *p* < 0.001). These results strongly suggest that patients with more preserved post-1st-EBP MR CSF flow dynamics, even within the context of SIH, are more likely to respond favorably to a single EBP. This indicates that CSF flow parameters are not only valuable in distinguishing SIH from HVs but also hold substantial utility in predicting treatment outcome and the likelihood of achieving EBP success.

Currently, clinical decisions regarding the need for repeated EBP are often based on symptom persistence, imaging findings, or indirect indicators of CSF leakage severity [32,33]. Recently, the development of clinical scoring systems such as the SIH-EBP score has shown promise in stratifying patients’ likelihood of response. Our study adds to this growing body of evidence by demonstrating that post-1st-EBP MR CSF flow parameters represent a robust physiological marker with complementary predictive value [34,35]. Higher post-1st-EBP MR CSF flow might reflect a less severe or more amenable CSF leak, or perhaps a greater residual capacity for the craniospinal system to restore normal dynamics after intervention. Further studies integrating these quantitative CSF flow parameters with anatomical leak characteristics and clinical presentations are warranted to develop more precise, individualized treatment algorithms and enhance the predictability of EBP success in SIH management.

Beyond its clinical insights, our study’s most significant academic contribution lies in the establishment of a novel AI-driven framework for quantitative CSF flow analysis, offering enhanced reliability and efficiency in diagnosing SIH. This methodology, which uniquely integrates YOLOv4 for initial CSF region detection with Independent Component Analysis (ICA) and Pulsatility-Based Segmentation (PUBS), represents a considerable advancement over existing approaches.

Traditional methods for CSF flow quantification, such as manual region-of-interest (ROI) delineation or simple intensity thresholding, are inherently labor-intensive, highly susceptible to inter-operator variability, and can significantly impact reproducibility [36]. Our integrated AI framework directly addresses these limitations. The initial YOLOv4 detection provides a robust and rapid method for identifying CSF spaces, minimizing the inclusion of non-CSF tissues. Subsequently, the application of ICA allows for the robust separation of the underlying CSF flow signal from confounding physiological noise and artifacts, a common challenge in phase-contrast MRI data [27]. Complementing this, PUBS leverages the intrinsic pulsatile nature of CSF flow to refine segmentation and extract flow parameters, adapting to individual patient dynamics without requiring external reference waveforms [28,37,38]. This yields a fully automated workflow that reduces manual segmentation time and variability, ensuring high reproducibility [28,36,39,40]. Building on the principles, our AI-driven uses pulsatility features directly from the phase-contrast data, making it more adaptable and robust than methods dependent on fixed reference waveforms and enabling rapid and consistent extraction of CSF flow parameters across consecutive scans [21,41]. This methodological advancement not only enhances the diagnostic utility of CSF flow metrics but also lays a critical groundwork for future individualized treatment planning by providing highly standardized and reliable quantitative data.

## 5. Limitations

This study has several limitations. The sample size was relatively small, and although follow-up MRI data were available for most patients, the follow-up period was limited in duration. Moreover, the classification of “recovered” patients relied on short-term clinical and imaging resolution, without the longitudinal tracking of outcomes. Future research should validate these findings in larger, prospectively enrolled cohorts with extended follow-up to assess symptom recurrence and the sustained response to EBP treatment. Additionally, integrating CSF flow parameters with anatomical characteristics of the leak may improve the predictive accuracy of models assessing EBP efficacy.

On the other hand, the nearly perfect AUC for the post-1st-EBP classification likely reflects the small sample (*n* = 25; 13 responders, 12 non-responders) and the limited overlap of the CSF flow metric between groups. Because thresholds were derived on the same dataset, overfitting and optimism are possible. These findings should be interpreted as hypothesis-generating and confirmed in larger, prospective cohorts with external/temporal validation. Accordingly, we will report bootstrap 95% CIs and cross-validated performance to quantify uncertainty and reduce optimism.

Future work should validate these observations in larger, prospectively enrolled, multi-center cohorts with external and temporal validation, ideally across different vendors and field strengths (1.5 T/3 T) to assess generalizability. Integrating cervical CSF-flow metrics with anatomical leak characteristics (leak type and level, SLEC status, CSF–venous fistula) and patient covariates (e.g., age, sex, blood pressure) may yield more robust prediction of first-EBP success versus first-EBP failure. Finally, test–retest repeatability and inter-reader robustness of the automated pipeline should be quantified, with code and parameter presets made available for reproducibility.

## 6. Conclusions

In conclusion, AI-enhanced quantitative phase-contrast MRI (PC-MRI) offers a robust and objective method for assessing cerebrospinal fluid (CSF) flow dynamics in patients with spontaneous intracranial hypotension (SIH). This technique not only facilitates accurate differentiation between SIH patients and HVs but also provides insights into the physiological reversibility of CSF flow impairment following treatment. Importantly, specific post-1st-EBP MR CSF flow parameters have demonstrated excellent predictive performance for identifying patients likely to respond favorably to EBP, with some metrics achieving near-perfect or perfect diagnostic accuracy. These findings highlight the significant clinical utility of quantitative imaging biomarkers in both diagnosis and treatment stratification, potentially guiding therapeutic decision-making. Future studies that integrate CSF flow dynamics with clinical and anatomical variables may further refine individualized treatment planning and enhance outcomes in the management of SIH.

## Figures and Tables

**Figure 1 diagnostics-15-02339-f001:**
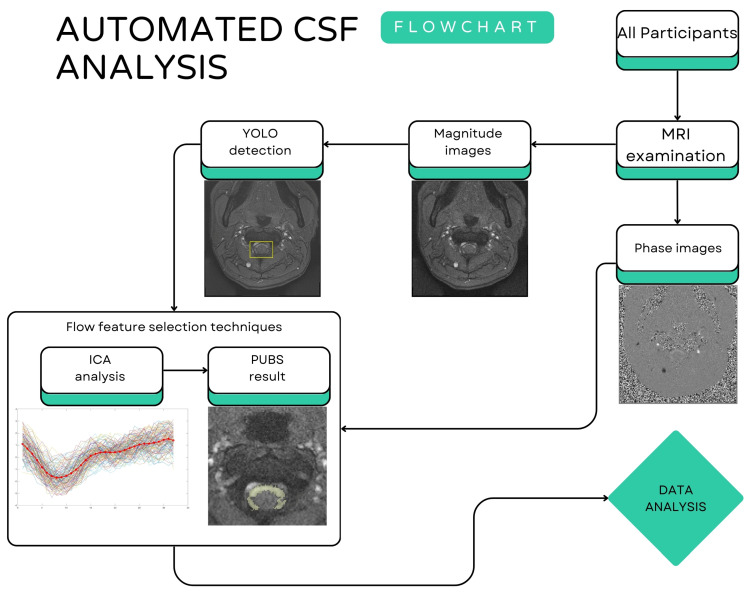
The schematic workflow of automated CSF analysis for all participants.

**Figure 2 diagnostics-15-02339-f002:**
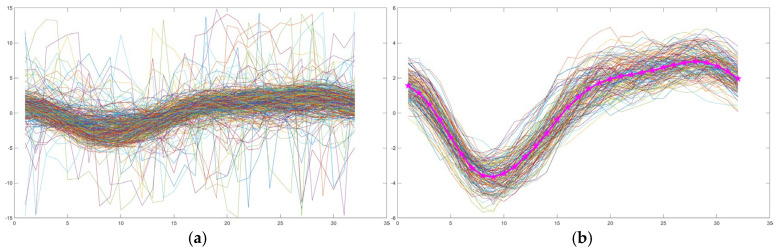
ICA flow signals from the spinal canal region (**a**). The line linked by pink asterisk is the preliminary CSF reference velocity waveform (**b**).

**Figure 3 diagnostics-15-02339-f003:**
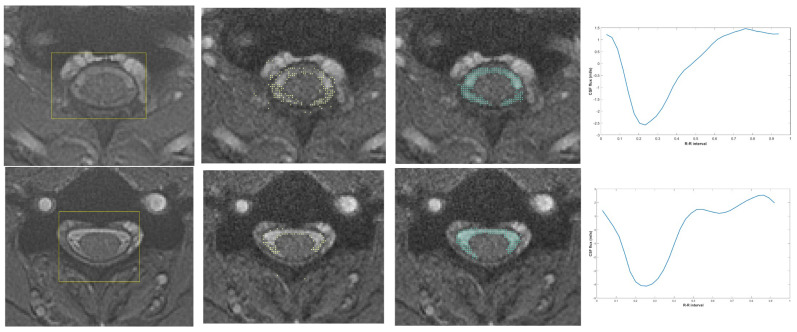
YOLOv4 was utilized for the detection of the spinal canal (the area within the yellow box), while ICA reference points (the yellow dots) were established to define the waveform. The PUBS algorithm thereafter effectively identified the flowing CSF areas (the green dots), enabling the analysis of pulsatile CSF flux within a single R-R interval.

**Table 1 diagnostics-15-02339-t001:** Demographic data of participants.

	SIH	HVs	*p*
Number of cases	31	26	
Sex (female/male)	22/9	15/11	0.575
Age (years old) (mean ± SD)	39.58 ± 9.99	38.12 ± 6.82	0.724
Days of Headache onset to the first MRI (range; median)	2–60; 8	Not applicable	
Epidural blood patch (EBP)	25	Not applicable	
Hydration only	6	Not applicable	
Times of EBP > 1	12	Not applicable	
Days of the first MRI to the first EBP (range; median)	1–7; 2	Not applicable	
Days of the first EBP to the follow-up post-1st-EBP MRI (range; median)	1–14; 2	Not applicable	

Mann–Whitney test. Chi-Square test.

**Table 2 diagnostics-15-02339-t002:** Comparison of CSF flow parameters at baseline MR between SIH patients and healthy volunteers.

Characteristic	Group		*p* Value
	SIH Patients (*n* = 31)	HVs (*n* = 26)	
	Mean ± SD	Mean ± SD	
Upward mean flow (mL/s)	0.76 ± 0.31	1.18 ± 0.34	<0.001 **
Downward mean flow (mL/s)	1.01 ± 0.43	1.60 ± 0.54	<0.001 **
Summation of mean flow (mL/s)	1.77 ± 0.72	2.78 ± 0.84	<0.001 **
Upward peak flow (mL/s)	1.28 ± 0.50	1.87 ± 0.52	<0.001 **
Downward peak flow (mL/s)	1.76 ± 0.77	2.96 ±0.92	<0.001**
Summation of peak flow (mL/s)	3.04 ± 1.26	4.83 ± 1.39	<0.001 **
Upward CSF total flow (mL/cycle)	14.32 ± 5.59	22.24 ± 6.55	<0.001 **
Downward CSF total flow (mL/cycle)	13.22 ± 5.65	20.74 ± 6.66	<0.001 **
Absolute stroke volume (mL/cycle)	27.54 ± 11.03	42.98 ± 12.86	<0.001 **

Mann–Whitney test, ** *p* < 0.01.

**Table 3 diagnostics-15-02339-t003:** Comparison of CSF flow parameters between complete recovered SIH patients and healthy volunteers.

Characteristic	Group		*p* Value
	Recovered SIH Patients (*n* = 24)	HVs (*n* = 26)	
	Mean ± SD	Mean ± SD	
Upward mean flow (mL/s)	1.29 ± 0.21	1.18 ± 0.34	0.193
Downward mean flow (mL/s)	1.84 ± 0.36	1.60 ± 0.54	0.068
Summation of mean flow (mL/s)	3.13 ± 0.52	2.78 ± 0.85	0.074
Upward peak flow (mL/s)	2.00 ± 0.32	1.87 ± 0.52	0.174
Downward peak flow (mL/s)	3.12 ± 0.52	2.96 ±0.92	0.294
Summation of peak flow (mL/s)	5.12 ± 0.78	4.83 ± 1.39	0.193
Upward CSF total flow (mL/cycle)	24.86 ± 4.86	22.24 ± 6.55	0.099
Downward CSF total flow (mL/cycle)	23.32 ± 3.56	20.74 ± 6.66	0.060
Absolute stroke volume (mL/cycle)	48.18 ± 7.22	42.98 ± 12.86	0.077

Mann–Whitney test.

**Table 4 diagnostics-15-02339-t004:** Diagnostic Performance of baseline PC-MRI CSF Flow Parameters for discriminating SIH from healthy volunteers.

Parameter	AUC	Cutoff Value	Sensitivity (%)	Specificity (%)
Upward mean flow (mL/s)	0.825	0.8426	67.7	84.6
Downward mean flow (mL/s)	0.813	1.2314	77.4	73.1
Summation of mean flow (mL/s)	0.828	1.9812	71.0	88.5
Upward peak flow (mL/s)	0.793	1.4647	71.0	88.5
Downward peak flow (mL/s)	0.844	2.0964	71.0	92.3
Summation of peak flow (mL/s)	0.841	3.5265	74.2	92.3
Upward CSF total flow (mL/cycle)	0.819	16.7697	74.2	80.8
Downward CSF total flow (mL/cycle)	0.819	15.7528	71.0	84.6
Absolute stroke volume (mL/cycle)	0.829	30.8367	71.0	88.5

AUC, sensitivity, specificity and optimal thresholds for distinguishing SIH from healthy volunteers.

**Table 5 diagnostics-15-02339-t005:** Post-1st-EBP MR CSF flow parameters between first EBP successful patients and first EBP failure patients.

Characteristic	Group		*p* Value
	EBP Failure Patients (*n* = 12)	EBP Successful Patients (*n* = 13)	
	Mean ± SD	Mean ± SD	
Upward mean flow (mL/s)	0.691 ± 0.14	1.26 ± 0.23	<0.001 **
Downward mean flow (mL/s)	0.84 ± 0.24	1.81 ± 0.46	<0.001 **
Summation of mean flow (mL/s)	1.53 ± 0.36	3.07 ± 0.65	<0.001 **
Upward peak flow (mL/s)	1.13 ± 0.24	2.00 ± 0.34	<0.001 **
Downward peak flow (mL/s)	1.45 ± 0.42	2.98 ± 0.64	<0.001 **
Summation of peak flow (mL/s)	2.58 ± 0.61	4.98 ± 0.95	<0.001**
Upward CSF total flow (mL/cycle)	12.47 ± 2.64	24.26 ± 5.13	<0.001 **
Downward CSF total flow (mL/cycle)	14.58 ± 2.95	22.92 ± 4.35	<0.001 **
Absolute stroke volume (mL/cycle)	24.04 ± 5.45	47.17 ± 9.16	<0.001 **

Values are derived from PC-MRI at C2 acquired after the first EBP. Groups: EBP success = recovery after a first EBP; EBP failure = patients need multiple EBP; Mann–Whitney test. ** *p* < 0.01.

**Table 6 diagnostics-15-02339-t006:** Diagnostic performance of post-1st-EBP CSF flow metrics for stratifying need for repeated EBP.

Parameter	AUC	Cutoff Value	Sensitivity (%)	Specificity (%)
Upward mean flow (mL/s)	0.994	0.902	92.3	100
Downward mean flow (mL/s)	1	1.2511	100	100
Summation of mean flow (mL/s)	0.994	2.1521	92.3	100
Upward peak flow (mL/s)	0.994	1.4923	92.3	100
Downward peak flow (mL/s)	1	2.1968	100	100
Summation of peak flow (mL/s)	1	3.4623	100	100
Upward CSF total flow (mL/cycle)	0.974	17.1383	92.3	100
Downward CSF total flow (mL/cycle)	1	16.4993	100	100
Absolute stroke volume (mL/cycle)	1	33.3892	100	100

## Data Availability

Data are available on request due to restrictions (e.g., privacy, legal or ethical reasons). The data presented in this study are available on request from the corresponding author due to IRB restriction.

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
