# Peer review of "AI-Augmented Quantitative MRI Predicts Spontaneous Intracranial Hypotension"

_diagnostics, 2025, doi:10.3390/diagnostics15182339_

Round 1

Reviewer 1 Report

Comments and Suggestions for Authors

Paper is about Spontaneous intracranial hypotension (SIH), caused by spinal cerebrospinal 18 fluid (CSF) leakage. These disease is very important for human life. Authors present YOLO based a classifcation method for MRI estimation. Method is good. But comparision table must be present and explained. 

Comments on the Quality of English Language

Language is good. 

Reviewer 2 Report

Comments and Suggestions for Authors

AI-Augmented Quantitative MRI Predicts Spontaneous Intracranial Hypotension

The authors present a very interesting application of an AI-supported evaluation of MRI images for the diagnosis of spontaneous intracranial hypotension. With their method, they achieve a high accuracy.

However, the descriptions of the methodology in particular are somewhat imprecise. I have the following comments:

Introduction: What are the advantages of focusing on cervical spine imaging? Is the diagnosis of the aqueduct not precise enough?

Materials and Methods

2.1 Study Design and Participants:

  • It is stated that patients were „retrospectively and prospectively“ enrolled. This is unusual. Please specify.
  • What were the inclusion criteria for the healthy volunteers? Were they prospectively recruited? Were age and gender the only matching criteria?

2.2 Imaging Techniques

  • „PC-MRI was performed at both the initial diagnosis and post-epidural blood patch.“ But some patients were hydrated only. Was a second MRI performed in these patients?
  • When was the second MRI imaging performed in the patients receiving more than one blood patch?

2.3 Artificial Intelligence-Based Flow Analysis

  • please include the MATLAB version in the text. The formal reference („Inc., 2024“) sounds unusual.
  • Please provide a description of Figure 1. What about the control patients?
  • It is stated that „A total of 151 independently collected images were used for training purposes“. Does this refer to pre-training of the algorithm? Later, it is stated that the test dataset consists of 46 cases. Where do these 46 cases come from? Did the 151 images come from the 46 patients mentioned?
  • Please describe the pre-training/fine-tuning in more detail and indicate which image material you used for this. Were the 46 patients used for pre-training different from the 31 examined and the 26 healthy controls?
  • Please describe the data augmentation methods (random horizontal flipping, scaling, and rotation) in more detail

2.4 Treatment and Follow-Up

  • How many lumbar, thoracic and cervical blood patches were applied? Was the site associated with EBP failure?

Results

3.1 Participant Characteristics

  • Table 1: In my opinion, „hydration only“ and „Times of EBP>1“ do not apply to the controls
  • as patients and controls were age- and sex-matched, there is no significant difference expected

3.3 CSF Flow Comparison After Recovery

  • Why did only 24 SIH patients have follow-up imaging? If there were 6 patients without EBP, at least 25 should have had follow-up imaging

3.5 Association Between CSF Flow Parameters and the Number of EBPs

  • It would rather be „Association between CSF Flow Parameters and EBP Failure“, as the absolute number of EPB attempts is not analysed.
  • Tables 4 and 5: Please correct „stoke volume“ to „stroke volume“

3.6 Diagnosit Performance of CSF Flow Metrics for Predicting EBP Effectiveness

  • AUC of 1.0, Sensitivity and specificity of 100% sounds a bit unrealistic. Please add the possibility of overfitting to the discussion. Was the dataset too small, or were the applied augmentation methods (random horizontal flipping, scaling, and rotation) not sufficient?

In the introduction, the authors state: „The findings of CSF flow changes at the upper cervical spinal level (C2/C3) remain inconsistent across studies for SIH patients.“ This sentence should be taken up again in the discussion.

Reviewer 3 Report

Comments and Suggestions for Authors

Thank you for inviting me to review this manuscript, where authors investigate the utility of cervical CSF flow metrics as novel, objective imaging biomarkers for the diagnosis of SIH. Authors did a case-control to assess the role of this AI augmented MRI, and show some interesting results. In my opinion, the study is interesting and very well prepared. All sections are clear and follow the guidelines. I believe that this manuscript would offer an addition to the literature. Authors could highlight some future research suggestions in the field. This would be my only comment.

Good luck

Round 2

Reviewer 2 Report

Comments and Suggestions for Authors

The authors have adequately revised the manuscript. I only have some minor remarks:

Two sentences are worded ambigously:

Lines 75-77

„In fact,  most CSF leaks in SIH arise from the spine, we asked whether site-proximal cervical (C2–C3) flow metrics could also reflect disease status and severity.“ Does this mean: „As most CSF leaks in SIH arise from the spine, we asked whether site-proximal cervical (C2–C3) flow metrics could also reflect disease status and severity.“ 

Lines 109-113

„Classical unsupervised clustering approaches (e.g., k-means, expectation–maximization) and adaptive thresholding methods, often combined with anatomical priors and connectivity analysis, remain in limited use because of their computational efficiency; however, their accuracy in complex pathological scenarios generally lags behind deep learning–based techniques.“ 

Does it mean: „Classical unsupervised clustering approaches […]  are computationally efficient; however, their accuracy in complex pathological scenarios generally lags behind deep learning–based techniques.“ 

Lines 186-188:

The protocol was approved by the institutional review board (SE17334A, CE16103A, CE22242B), and written informed consent was obtained in accordance with the approved procedures.

Please add that informed consent was obtained by the healthy volunteers (not by the retrospective studied cohort for this special study).

Lines 234-235

 (The MathWorks Inc. (2024). MATLAB version: 24.1.0.2603908 (R2024a)) (Inc., 2024). Please remove the last „(Inc., 2024)“.

Figure 1:

As there is no difference between SIH patients and all participants, the figure could be reduced to the first part.

Lines 341-342

„For each patient, the vertebral level of the first EBP was recorded and categorised as cervical, thoracic, or lumbar. (Table S3)“

Please add in the text that there were no cervical blood patches.

Table 4: Please clarify in the subscript that the table shows the diagnostic performance for discrimination of healthy and SIH.

Lines 413-414: „3.5. Association Between post-1st-EBP PC-MRI CSF Flow Parameters and the Number of EBPs (Table 5)“ This must be the subscript for Table 5, which is redundant to lines 434-435,

lines 438-439: subscript for table 6, redundant to lines 456-457

References: There is a missing (deleted) reference 36. Please correct the numbering of the following references. Reference 41 (van Pelt) is not cited in the text.

Comments on the Quality of English Language

I have no remarks except of lines 75-77 and lines 109-113 which should be rephrased
